# Radical Hysterectomy for Early Stage Cervical Cancer

**DOI:** 10.3390/ijerph191811641

**Published:** 2022-09-15

**Authors:** Giorgio Bogani, Violante Di Donato, Giovanni Scambia, Francesco Raspagliesi, Vito Chiantera, Giulio Sozzi, Tullio Golia D’Augè, Ludovico Muzii, Pierluigi Benedetti Panici, Ottavia D’Oria, Enrico Vizza, Andrea Giannini

**Affiliations:** 1Department of Maternal and Child Health and Urological Sciences, Sapienza University of Rome, Policlinico Umberto I, 00185 Rome, Italy; 2Gynecologic Oncology Unit, Fondazione Policlinico Universitario A. Gemelli IRCCS, 00182 Rome, Italy; 3Gynecologic Oncology Unit, Fondazione IRCCS Istituto Nazionale dei Tumori di Milano, 20133 Milan, Italy; 4Department of Gynecologic Oncology, University of Palermo, 90127 Palermo, Italy; 5Gynecologic Oncology Unit, Department of Experimental Clinical Oncology, IRCCS “Regina Elena” National Cancer Institute, 00144 Rome, Italy

**Keywords:** cervical cancer, LACC, laparoscopy, robotic, radical hysterectomy

## Abstract

Radical hysterectomy and plus pelvic node dissection are the primary methods of treatment for patients with early stage cervical cancer. During the last decade, growing evidence has supported the adoption of a minimally invasive approach. Retrospective data suggested that minimally invasive surgery improves perioperative outcomes, without neglecting long-term oncologic outcomes. In 2018, the guidelines from the European Society of Gynaecological Oncology stated that a “minimally invasive approach is favored” in comparison with open surgery. However, the phase III, randomized Laparoscopic Approach to Cervical Cancer (LACC) trial questioned the safety of the minimally invasive approach. The LACC trial highlighted that the execution of minimally invasive radical hysterectomy correlates with an increased risk of recurrence and death. After its publication, other retrospective studies investigated this issue, with differing results. Recent evidence suggested that robotic-assisted surgery is not associated with an increased risk of worse oncologic outcomes. The phase III randomized Robotic-assisted Approach to Cervical Cancer (RACC) and the Robotic Versus Open Hysterectomy Surgery in Cervix Cancer (ROCC) trials will clarify the pros and cons of performing a robotic-assisted radical hysterectomy (with tumor containment before colpotomy) in early stage cervical cancer.

## 1. Introduction: Before the Laparoscopic Approach to Cervical Cancer (LACC) Trial

Cervical cancer represents one of the most common gynecological malignancies worldwide. Despite the strong implementation of primary and secondary prevention (especially in developed countries), cervical cancer is still a source of concern. Cervical cancer is the fourth most common cancer among women globally, with an estimated 14,000 new cases and 4000 cervical-cancer-related deaths in the United States in 2022 [1].

Surgery is the primary method of treatment in early stage cervical cancer [2]. Radical hysterectomy allows tumor removal and identification of risk factors for tailor adjuvant treatments [3]. Accumulating data have highlighted the safety and effectiveness of radical hysterectomy (plus pelvic node dissection) in early stage cervical cancer [3,4,5]. Hence, the International Federation of Gynecology and Obstetrics (FIGO) recommended the utilization of radical hysterectomy [6]. The National Comprehensive Cancer Network (NCCN) in 2017 and the European Society of Gynaecological Oncology/European Society for Radiotherapy and Oncology/European Society of Pathology (ESGO/ESTRO/ESP) in 2018 recommended the execution of radical hysterectomy via open or minimally invasive surgery [7,8]. Until 2018, several retrospective studies had highlighted the safety and effectiveness of minimally invasive radical hysterectomy [9,10]. Owing to this mounting data, the scientific community strongly supported the use of minimally invasive radical hysterectomy [7,8,9,10,11,12]. Wang et al. reported data from another meta-analysis on 12 studies collecting data on 754 and 785 patients who underwent laparoscopic and open radical hysterectomy, respectively [11]. This meta-analysis showed that disease-free and overall survival were similar between the two approaches [11]. In another meta-analysis, data from 2922 patients undergoing surgery for cervical cancer (1230 and 1692 undergoing laparoscopic and open surgery, respectively) corroborated the previous findings. The pooled data suggested that the survival results were similar between patients treated with laparoscopic and open radical hysterectomy [11]. As a result, the ESGO/ESTRO/ESP guidelines stated that a “Minimally invasive approach is favored (grade B)” [7]. However, no level A existed. To bridge this gap, the Laparoscopic Approach to Cervical Cancer (LACC) trial was designed to test the merit of minimally invasive radical hysterectomy compared with the open approach [13].

## 2. The Laparoscopic Approach to Cervical Cancer (LACC) Trial

In 2018, Ramirez et al. published the results of the prospective, randomized, phase III LACC Trial [13]. The LACC trial was stopped early (631 eligible patients had been enrolled of the initially planned 740 patients) due to its unexpected results [13]. The trial included patients with stage IA1 (with lymphovascular invasion), IA2, or IB1 cervical cancer. Only patients with squamous cell cervical cancer, adenocarcinoma, and adenosquamous carcinoma were included. Owing to the nature of the study design (prospective, randomized trial), the groups were homogeneous. The LACC trial included 319 patients in the minimally invasive group. Among those patients, 84.4% and 15.6% had laparoscopic and robotic-assisted surgery, respectively. The main outcome measure of the LACC trial was the rate of disease-free survival at 4.5 years. Ramirez et al. observed that patients who underwent minimally invasive surgery experienced worse 4.5-year survival outcomes in comparison with patients who had open radical hysterectomy [13]. In particular, the rate of disease-free survival was 86% following minimally invasive radical hysterectomy and 96.5% following open abdominal radical hysterectomy. Patients undergoing minimally invasive radical hysterectomy were more likely to develop a loco-regional recurrence (HR: 4.26, 95%CI: 1.44, 12.60). The 3-year disease-free survival following minimally invasive and open radical hysterectomy was 91.2% and 97.1%, respectively. Similarly, the 3-year disease-free survival following minimally invasive and open radical hysterectomy was 93.8% and 99.0%, respectively [13]. These findings provided a deep paradigm shift in gynecologic oncology practice [14].

Although the LACC trial was not sufficiently powered to enable subanalysis based on the stage of the disease, the authors observed that the surgical approach only influenced the risk of recurrence for patients with a tumor larger than 2 cm; while for patients with stage IB < 2 cm, the execution of minimally invasive radical hysterectomy did not impact the patient’s prognosis (1/147 patients in the open group vs. 5/150 patients in the minimally invasive group; *p* = 0.214). Additionally, two other studies evaluated the outcomes of patients included in the randomized LACC trial [15,16]. Obermair et al. evaluated the incidence of adverse events (including intraoperative and postoperative events within 6 months after surgery) in patients who received minimally invasive and open radical hysterectomy [15]. Frumovitz et al. evaluated the quality of life (at baseline, -weeks, and 3 months after surgery) in the LACC population [16]. These studies highlighted that patients undergoing minimally invasive surgery had a similar risk of treatment-related adverse events to patients undergoing open surgery. Moreover, quality of life (assessed using validated quality-of-life and symptom assessments) was not influenced by the type of surgical approach [16]. Based on these data, patients with early stage cervical cancer undergoing minimally invasive surgery are at high risk of recurrence [13]. Additionally, in comparison with open surgery, the execution of minimally invasive surgery does not correlate with improved perioperative outcomes, in terms of morbidity rate or quality of life [15,16].

## 3. After the LACC Trail

The LACC trial was not the only study observing the detrimental effect of minimally invasive radical hysterectomy [17]. Melamed et al. reported data from a cohort study evaluating how the use of minimally invasive radical hysterectomy impacted survival. The authors performed two different analyses, using data from (i) the Commission on Cancer-accredited hospitals in the United States (2010–2013); (ii) the Surveillance, Epidemiology, and End Results (SEER) program database (2000–2010) [17]. The primary analysis included 2461 patients (1225 undergoing minimally invasive radical hysterectomy). Patients who underwent minimally invasive surgery were more likely to have smaller and lower-grade tumors. The authors adopted an inverse probability weighting analysis to correct confounding factors. The 4-year mortality was 9.1% and 5.3% for patients having a minimally invasive and open radical hysterectomy, respectively. In the second analysis, the authors observed that the adoption of minimally invasive surgery coincided with a decrease in the survival rate of 0.8% per year [17].

As abovementioned, two meta-analyses observed that minimally invasive and open surgery result in similar oncologic outcomes [11,12]. However, these meta-analyses did not include recent research in this field. In 2020, Nitecki et al. conducted a new meta-analysis on this issue, involving 9499 patients [18]. This new meta-analysis corroborated the findings of the LACC trial [13]. The pooled hazard ratio was 71% higher for patients having minimally invasive surgery in comparison with patients having open radical hysterectomy [18]. After the publication of the results of LACC trial, other studies observed similar findings, even in the population with tumor diameter < 2 cm [19,20]. However, another randomized prospective study (a part of the LACC trial [13]) compared minimally invasive and open radical hysterectomy [20]. This study enrolled only 30 patients (including 16 and 14 laparoscopic and open radical hysterectomy, respectively).

The overall survival hazard ratio was 2.05 (CI 95%: 0.51–8.24), and the disease-free survival hazard ratio was 2.13 (CI 95%: 0.39–11.7). The difference was not statistically significant (possibly due to the small sample size) [20].

The publication of the results of the LACC trial provided important changes in the management of cervical cancer patients worldwide. The Society of Gynecologic Oncology (SGO) encouraged surgeons to discuss with patients the data of the LACC trial, thus proving adequate counseling [21]. The NCCN suggested that open surgery should be the “standard and recommended approach to radical hysterectomy” [22]. After the publication of the results of the LACC trial, the number of procedures performed via laparoscopic and robotic-assisted surgery dramatically decreased. Lewicki et al. reported the effects of the publication of the results of the LACC trial on clinical practice in the United States [23]. Using data from the Premier Healthcare Database, the authors observed a decrease in the adoption of minimally invasive radical hysterectomy (from 58% to 42.9%) [23]. This decrease was much more evident in academic centers (73%) than in nonacademic ones (13%) [23].

All the studies focusing on practice changes have highlighted an important decrease in the adoption of a minimally invasive approach [24,25,26]. However, these studies disagree in the implication of this paradigm shift [24,25,26]. Some investigations suggested that the increased adoption of open radical hysterectomy correlates with an increased risk of postoperative events [24,25]. An analysis of the National Inpatient Sample from October 2015 to December 2018 suggested that after the publication of the results of the LACC trial, cervical cancer patients were less likely to have minimally invasive radical hysterectomy (−63%), but more likely to develop perioperative complications (+23%) and longer length of hospital stay (3 vs. 2 days) [24]. Similarly, a recent study from the Italian Gynecological Cancer Study group observed that the adoption of minimally invasive radical hysterectomy decreased after the publication of the LACC trial results from 64.9% to 30.4% (*p* < 0.001). However, the morbidity rate was not influenced. Overall 90-day complications occurred in 110 (18.9%) and 119 (16.6%) patients before and after the publication of the LACC trial results, respectively (*p* = 0.795). Similarly, the number of severe (grade 3 or worse) complications did not differ between the two periods (38 (6.5%) vs. 37 (5.1%); *p* = 0.297). More interestingly, overall and severe 90-day complications were consistent between periods, even in stages IA (*p* = 0.471), IB1 (*p* = 0.929), and IB2 (*p* = 0.074), separately [26]. These data supported the findings observed in the LACC trial [13,14,15]. Further evidence is needed to understand the pros and cons of adopting one approach rather than the other in terms of morbidity rate and quality of life.

The exact reasons why the minimally invasive approach correlates with worse disease-free and overall survival are unknown. Some experts identified three possible reasons: (i) lack of radicality, (ii) surgeon expertise, and (iii) tumor dissemination at the time of colpotomy [27,28]. The latter is the most credible reason. Klapdor et al. performed a proof-of-principle study. Indocyanine-green (IGC) was applied to the uterine cervix surface before the execution of laparoscopic and robotic-assisted hysterectomy. Peritoneal contamination and instrument contamination (with IGC) occurred in 75% and 60% of patients, respectively [27]. For these reasons, accumulating data highlighted that patients undergoing tumor removal (thorough conization) before surgery experience improved outcomes [29,30,31]. These studies supported that primary conization might overcome the risk of local recurrence after laparoscopic radical hysterectomy in early stage cervical cancer. The pattern of recurrence is another proof of concept of the possible tumor dissemination at the time of colpotomy [32]. Patients undergoing laparoscopic radical hysterectomy are more likely to experience intrapelvic recurrence and develop peritoneal carcinomatosis in comparison with patients undergoing open surgery [32,33].

The LACC trial produced a paradigm shift after its publication [14]. Other retrospective experiences corroborated the detrimental effects of adopting minimally invasive surgery [19,34]. The European cohort observational study comparing minimally invasive versus open abdominal radical hysterectomy in patients with stage IB1 cervical cancer (the SUCCOR study) included 1272 patients treated in 2013 and 2014. Overall, 693 patients met the inclusion criteria: 291 and 402 patients had a minimally invasive and open radical hysterectomy, respectively. Among patients in the minimally invasive surgery group, 228 (78.5%) underwent laparoscopic surgery and 63 (21.5%) underwent robotics surgery [34]. Patients undergoing minimally invasive radical hysterectomy experienced a 2- and 2.4-fold increase in the risk of recurrence and death, respectively [34]. However, recent studies reported contradictory results [35,36,37]. The CIRCOL group study reported outcomes of a Brazilian cohort who underwent minimally invasive vs. open radical hysterectomy in early stage cervical cancer. The three-year disease-free survival was 88.2% and 90.3% after minimally invasive and open surgery, respectively. Similarly, the five-year overall survival was 91.8% and 91.1%, respectively. Tumor diameter did not influence disease-free and overall survival [36]. More recently, the MEMORY study showed similar findings, analyzing a cohort of 1093 patients treated between 2007 and 2016. The study included: 715 minimally invasive radical hysterectomies (including 558 (78%) robotic-assisted procedures) and 378 open radical hysterectomies [38]. The three-year progression-free survival was 87.9% and 89% for minimally invasive and open procedures, respectively. Similarly, the three-year overall survival rate was 95.8% and 96.6%, respectively [38]. The MEMORY study had a particularly high proportion of patients who underwent robotic-assisted procedures [38]. The LACC trial, the SUCCOR study, and other studies were evaluated for most patients who had laparoscopic surgery [13,20,34]. A notable criticism against the LACC trial is the low prevalence of robotic-assisted radical hysterectomy [13]. Two ongoing trials are evaluating the role of robotic-assisted surgery in early stage cervical cancer. The phase III prospective Robotic-assisted Approach to Cervical Cancer (RACC) trial aims to enroll 800 patients, randomized to undergo robotic radical hysterectomy vs. open radical hysterectomy. Progression-free survival is the primary endpoint [39]. The trial of Robotic Versus Open Hysterectomy Surgery in Cervix Cancer (ROCC) is a multicenter, open-label, randomized, noninferiority clinical trial with the hypothesis that robotically assisted radical hysterectomy with tumor containment before colpotomy is noninferior to abdominal radical hysterectomy concerning disease-free survival. Overall, 840 patients will be included [39]. These trials will clarify the impact of robotic-assisted surgery and tumor containment in patients having a radical hysterectomy.

Another important point that deserves discussion is the role of a fertility-sparing approach in early stage cervical cancer [40]. Growing evidence supports the execution of conization plus node dissection in a large proportion of women affected by low-volume cervical tumors [41,42,43]. This approach must be considered during the counseling of women who wish to preserve their childbearing potential.

## 4. Conclusions

In the present research, we reviewed current evidence on the role of minimally invasive radical hysterectomy in early stage cervical cancer. To date, two randomized trials compared minimally invasive with open radical hysterectomy [13,20]. One of the trials included only 30 patients and was not able to assess differences in survival between the two approaches. However, the LACC trial clearly showed that minimally invasive radical hysterectomy is associated with worse survival. Based on this level A evidence, a minimally invasive approach should only be offered in the context of controlled trials. Further evidence from well-designed retrospective studies is warranted to improve knowledge on the treatment of early stage cervical cancer.

## Data Availability

Not applicable.

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
