# Peer review of "Radical Hysterectomy for Early Stage Cervical Cancer"

_ijerph, 2022, doi:10.3390/ijerph191811641_

Round 1

Reviewer 1 Report

The title of the report doesnot corespond fully with its contence . It is too general . The text may be subdidided in three parts: pre Ramirez period , Ramirez data, and post Ramirez findingsThe latter may describe explanation, and suggested methods which may improve oncological results of minnimaly invasive surgeries

Author Response

I would like to thank the reviewer for the comments. 

In order to comply with the reviewer's comment we subdivided the paper into 3 groups: pre-LACC trial, The data of the LACC trial, and post LACC trial. 

Reviewer 2 Report

Thanks for the nice paper. It gives an comprehensive and detailed overview of the review of comparing minimally invasive approach with open surgery, which have beneficial oncologic outcomes for patients with early-stage cervical cancer. The author not only compared the oncologic outcomes for patients, as well as perioperative complications and longer length of hospital stay. Meanwhile, Protective maneuvers for radical hysterectomy and recent evidence and ongoing trials were also discussed. Recommended for publication.

Author Response

We would like to thank the reviewer for this comment. 

No changes are requierd. 

Reviewer 3 Report

Thank you for allowing me to review this paper. this review reported data on available evidence regarding the role of MIS in managing cervical cancer.

The paper is well written. The language is OK for me

The data reported are exciting and deserve to be disseminated. 

I also appreciate the synthesis of the paper. In fact, the authors avoid adding useful table/figures (I agree that they are not needed for an article like this)

Interestingly, as reported in the paper there are a significant proportion of women who are still receiving MIS. This paper would be useful in disseminating data supporting the worse effect of adopting MIS in this context. 

No changes are required for me

Author Response

We thank the reviewer's for this comment. No changes are required. 

Reviewer 4 Report

 Review:

Thank you for the opportunity to review this manuscript. I find it particularly valuable. I have nothing to add, the only thing which I consider interesting to comment is what can you say about the age of the patients and there fertility. How do you make the decision when she has no children? Do you recommend ovarian biopsy with cryogen, then surrogate mother? Or do you prefer the conization first?

Author Response

We thank the reviewer for these comments. 

In order to comply with the reviewer's comment we added data regarding fertility -sparing.